# Development of Clinical-Grade Durvalumab-680LT and Nivolumab-800CW for Multispectral Fluorescent Imaging of the PD-1/PD-L1 Axis of the Immune Checkpoint Pathway

**DOI:** 10.3390/ph18101501

**Published:** 2025-10-07

**Authors:** Henrik K. Huizinga, Wouter T. R. Hooghiemstra, Matthijs D. Linssen, Derk P. Allersma, Bahez Gareb, Bart G. J. Dekkers, Wouter B. Nagengast, Marjolijn N. Lub-de Hooge

**Affiliations:** 1Department of Gastroenterology and Hepatology, University of Groningen, University Medical Center Groningen, 9713 GZ Groningen, The Netherlands; h.k.huizinga@umcg.nl (H.K.H.); wouterhooghiemstra@gmail.com (W.T.R.H.); w.b.nagengast@umcg.nl (W.B.N.); 2Department of Clinical Pharmacy and Pharmacology, University of Groningen, University Medical Center Groningen, 9713 GZ Groningen, The Netherlands; d_allersma@hotmail.com (D.P.A.); b.gareb01@umcg.nl (B.G.); b.g.j.dekkers@umcg.nl (B.G.J.D.); 3BioTherapeutics Unit, Division of Pharmacy and Pharmacology, Netherlands Cancer Institute, 1066 CX Amsterdam, The Netherlands

**Keywords:** fluorescent tracer development, fluorescent molecular imaging, multispectral imaging, immune checkpoint inhibitors, durvalumab, nivolumab, IRDye 680LT, toxicity

## Abstract

**Background:** Immune checkpoint inhibitors (ICIs) are effective against various advanced and metastatic cancers, but patient responses vary and can change over time, complicating treatment prediction. Therefore, better tools for patient stratification, response prediction, and response assessment are needed. This study presents the development and clinical translation of a fluorescently labelled ICI tracer pair used to perform multispectral fluorescent molecular imaging and simultaneously gain spatial and temporal insight in both programmed death ligand 1 (PD-L1) and programmed death receptor 1 (PD-1) expression. **Methods:** We conjugated the anti-PD-L1 antibody durvalumab to IRDye 680LT and the anti-PD-1 antibody nivolumab to IRDye 800CW. Tracers were developed and optimized for conjugation efficiency and purity to allow use in clinical trials. Stability was tested up to 12 months. An extended single-dose toxicity study in mice was performed for durvalumab-680LT and the unconjugated IRDye 680LT to demonstrate safety for first-in-human administration. **Results:** Durvalumab-680LT and nivolumab-800CW were successfully conjugated and purified. Conjugation optimization resulted in a robust production with labelling efficiencies of ≥88%. Long-term stability study of both tracers showed all parameters within end of shelf-life specifications for at least 12 months at 2–8 °C. No toxic effects were observed in doses up to 1000x the intended human dose for both IRDye 680LT and durvalumab-680LT, which are therefore considered safe for first-in-human use. **Conclusions:** We succeeded in the development and clinical translation of two novel fluorescent ICI tracers, durvalumab-680LT and nivolumab-800CW. Moreover, we demonstrated for the first time the safety of IRDye 680LT and durvalumab-680LT, enabling first-in-human use. Together, this makes durvalumab-680LT and nivolumab-800CW suitable for phase I/II clinical trials.

## 1. Introduction

Immune checkpoints play an essential part in the regulation of the immune system, protecting tissue from collateral damage during inflammation and preventing auto-immune reactions. One protein involved in immune checkpoint signalling is the programmed death receptor 1 (PD-1), which is expressed on activated T-cells. Binding of PD-1 to its ligand, programmed death receptor ligand 1 (PD-L1), results in the downregulation of the T-cell response and inhibition of the immune response. However, many cancer types exploit this response by upregulation of PD-L1 expression on tumour cells, avoiding T-cell mediated anti-tumour responses. This mechanism of immune evasion is targeted by treatment regimens that include immune checkpoint inhibitor antibodies (ICIs) targeting PD-1 or PD-L1, such as nivolumab and durvalumab [1,2,3,4].

Durvalumab and nivolumab have both been registered by the European Medicines Agency. After initial registration, the indications for these ICIs have been expanded, and recent studies have shown their efficacy in several forms of advanced and metastatic cancers, primarily non-small cell lung carcinoma (NSCLC) and urothelial carcinoma for durvalumab and NSCLC, renal cell carcinoma, Hodgkin lymphoma, urothelial carcinoma, and squamous cell carcinoma of the head and neck for nivolumab. Both ICIs have also been shown to provide a survival advantage over and generally fewer high-grade adverse events when compared to conventional chemotherapies in urothelial cell carcinoma and NSCLC [5,6,7]. However, response rates of these ICIs have been shown to vary between patients (10–70%, depending on type of disease) and can change over time, making it challenging to predict which patients will benefit from treatment [8,9,10,11]. Therefore, better tools for patient stratification, early response prediction, and response assessment are desirable to be able to select patients likely to benefit from ICI therapy. This may improve the effectiveness of cancer treatment, prevent unnecessary treatment, limit unwanted side-effects, and reduce medical costs [12].

Currently, patient selection for ICI therapy is performed mainly by the assessment of PD-L1 expression by immunohistochemistry (IHC) in biopsy specimens. However, accuracy is low due to several factors, including the size of the tissue specimens and heterogeneity of the tumour. This can lead to PD-L1 status misclassification in up to 52% of cases, which can be an issue for the administration of certain ICIs, as a positive IHC test is required for some indications like NSCLC [13,14,15]. Moreover, not all patients with high PD-L1-expressing tumours (on IHC) showed benefits from checkpoint inhibition therapy, while conversely, patients with negative PD-L1 expression showed response to anti PD-L1 treatment in several cases [16,17]. In contrast, small studies have shown that tumour uptake of ^89^Zr-atezolizumab, ^89^Zr-nivolumab, and ^89^Zr-pembrolizumab provided a more consistent positive correlation between tracer uptake and therapy response [18,19,20,21] when compared to IHC tissue staining. However, positron emission tomography (PET) imaging modalities have limitations, such as poor spatiotemporal resolutions and the inability for real-time imaging during endoscopy or surgery.

In contrast, fluorescent molecular imaging (FMI) can be used for real-time identification of tumours and tumour margins in oncologic surgery and endoscopy. This has previously been demonstrated in surgical interventions for, among others, breast cancer, peritoneal carcinomatosis, squamous cell head and neck carcinoma, and oesophageal adenocarcinoma [22,23,24,25]. Contrary to their radiolabelled counterparts, tissue samples of patients who have received fluorescent tracers can be stored after the procedure and can be analysed long after the procedure using ex vivo camera systems or fluorescence flatbed scanners, while fluorescent microscopy can be used to identify the cell types targeted by the tracer [26].

We hypothesize that FMI can also be performed using fluorescently labelled ICI antibodies, enabling real-time prediction and evaluation of ICI therapy response. This is a different approach to FMI than used in previous oncologic FMI studies with solid tumours and their markers, where tumour identification, margin assessment, and surgical resection were the main goals. We aim to achieve this novel approach to FMI by performing multispectral fluorescent imaging using two antibodies, one targeting PD-1 and the other targeting PD-L1, each conjugated to different near-infrared dyes with a distinct fluorescent spectrum; IRDye 800CW, the main dye used up until now, and the novel IRDye 680LT. This combination of dyes enables multispectral imaging due to their distinctive absorbance and emission spectra. IRDye 800CW has an absorption maximum at 774 nm and an emission maximum at 789 nm in PBS, whereas IRDye 680LT has an absorption maximum at 676 nm and an emission maximum at 693 nm in PBS. Multispectral imaging will be further enhanced by recent technological advancements in needle-based ultrasound-guided quantitative fluorescence molecular imaging, enabling fluorescence measurements in the whole body [27]. With this pair of tracers and this new technology, we aim to visualize both immune cells and tumour cells at both sides of the immune checkpoint pathway at the same time, getting more mechanistic insight into drug distribution of both therapeutic antibodies, the penetration of effector cells into the tumour microenvironment and interactions at a molecular immunological level within the tumour.

In this paper we describe the development process and clinical translation of such a tracer pair, consisting of durvalumab, conjugated to IRDye 680LT, and nivolumab, conjugated to IRDye 800CW. Moreover, we show the results of an extended single-dose toxicity study for both the unconjugated IRDye 680LT and the antibody–dye conjugate durvalumab-680LT, which have not seen in-human use previously.

## 2. Results

### 2.1. Conjugation, SE-HPLC Characterization, and ELISA

We successfully conjugated both nivolumab and durvalumab to both dyes. Fluorescence was confirmed at, respectively, 775 nm and 676 nm for the main protein peaks on the size-exclusion high-performance liquid chromatography (SE-HPLC) chromatograms for all tracers. SE-HPLC chromatograms at 280 nm were comparable to unconjugated antibody standard with regards to peak shape and retention time (Figure 1A, data only shown for durvalumab-680LT and nivolumab-800CW). Yield was ≥90% and label efficiency ≥77% for all tracers (Table 1). After purification of the labelled protein, aggregates were not detected and unconjugated dye was less than 0.50% for all tracers, except for nivolumab-680LT. As shown in Table 1, results for durvalumab-680LT and nivolumab-800CW were superior to their respective counterparts. Therefore, labelling optimization was performed only for durvalumab-680LT and nivolumab-800CW.

An indirect enzyme-linked immunosorbent assay (ELISA) was set up for both of these tracers to determine the binding affinity of durvalumab-680LT and nivolumab-800CW to their respective target. Figure 1B–C shows initial results as sigmoid curves of both tracers with their respective standard, the unmodified antibodies.

### 2.2. Process and Labelling Optimization

After initial labelling experiments, optimization of the labelling process was performed for durvalumab-680LT and nivolumab-800CW to start translation for clinical use. Experiments were performed at each step of the conjugation process to create a robust production process resulting in a stable tracer.

Labelling efficiency of durvalumab-680LT without purification resulted in half of the labelling efficiency with purification (38% versus 84%, respectively). Durvalumab was therefore purified before conjugation, as the original buffer contains interfering components, primarily amino acids [28] (Appendix A). No effect on the labelling efficiency was found with or without purification before the labelling, as the product buffer for nivolumab does not contain interfering components [29] (Appendix A). However, nivolumab was purified before labelling to ensure optimal labelling conditions. For both durvalumab-680LT and nivolumab-800CW, peak label efficiency was achieved after 20 min, which stayed stable for the full labelling period of 120 min (Figure 2A). To ensure complete labelling and to account for analysis time, labelling time was set to 60–120 min.

The chromatography at higher label ratios yielded broader peaks as the label ratio increased. Label efficiency peaked at a molar dye to protein ratio of 2:1 for both tracers, after which it declined (Figure 2B). Free dye and aggregates also increased at labelling ratios higher than 2:1 (Appendix A). The fluorescence gained by higher label ratios also rapidly declined relative to the amount of dye conjugated. Fluorescence even began to diminish when labelled at a ratio of 16:1 (Figure 2C). Therefore, to ensure sufficient fluorescence in vivo, but also ensure long term stability, the molar dye to protein ratio of 2:1 was chosen for both tracers.

Using the described labelling process, a buffer panel and small-scale stability study were performed for both tracers. Stability results for 28 days for the optimal buffer for both tracers, a 50 mM sodium phosphate buffer of pH 7.0, are shown in Figure 2D–F. For both tracers, all parameters stayed within end of shelf-life specifications for at the duration of the stability study, with concentrations between 0.99 and 1.06 mg/mL for durvalumab-680LT and 0.99 to 1.02 mg/mL for nivolumab-800CW. Percentages of free dye stayed between 0.6 and 2.8% for durvalumab-680LT and 0 and 1.3% for nivolumab-800CW. No aggregate formation was observed during the stability study of nivolumab-800CW, while aggregates stayed between 0 and 0.2% for durvalumab-680LT. Results for the other tested buffers are shown in Appendix A.

### 2.3. cGMP Production, Validation, and Long-Term Stability

For both tracers, a validation batch at scale for clinical trial manufacturing was successfully produced in our current good manufacturing practice (cGMP) facility. The produced vials were all included into a long-term stability study at 2–8 °C and 15–25 °C. Table 2 provides the product specifications at release and at end of shelf life and their methods for testing. Figure 3 shows the results obtained after the first 12 months of the stability study for both tracers at 2–8 °C. Complete quality control (QC) data can be found in Appendix A. The protein concentration, the percentage of free dye and aggregates, and the target binding affinity complied with the release specifications and remained within the end of shelf-life specification for both products during the first 12 months of the stability study. Concentration after production was 1.03 ± 0.01 mg/mL for durvalumab-680LT and was 1.07 ± 0.01 mg/mL after 12 months of stability testing, with a high of 1.09 ± 0.006 mg/mL after 3 months and low of 1.01 ± 0.01 mg/mL after 6 months. Percentage of free dye was 1.3 ± 0.1% on release for durvalumab-680LT and was 1.7 ± 0.06% after 12 months of stability testing, with a high at 6 months of 1.9 ± 0.1%. No measurement was lower during the stability study than the first measurement of free dye. No aggregate formation was observed during the stability study of durvalumab-680LT. Target binding affinity stayed between 67 and 98% for durvalumab-680LT. Concentration after production was 0.98 ± 0.003 mg/mL for nivoumab-800CW and was 0.97 ± 0 mg/mL after 12 months of stability testing, with a high of 1.01 ± 0.006 mg/mL after 3 months and low of 0.97 ± 0.003 mg/mL after 6 months. Percentage of free dye was 0.4 ± 0.09% on release for nivolumab-800CW and was 2.6 ± 0.15% after 12 months of stability testing. All measurements stayed within this range for the free dye. Aggregate formation was only observed for nivolumab-800CW after 12 months of stability testing with 0.7 ± 0.15% of aggregates. Target binding affinity stayed between 60 and 92% for nivolumab-800CW. Consistently, all other QC tests stayed within the end of shelf-life specifications for both tracers.

Stability data obtained during the first 12 months at 15–25 °C are presented in Appendix A, while complete QC data can be found in Appendix A. Both tracers also stayed within end of shelf-life specifications when stored at room temperature. Concentration after production was 1.03 ± 0.01 mg/mL for durvalumab-680LT and was 1.04 ± 0.006 mg/mL after 12 months of stability testing, with a high of 1.06 ± 0 mg/mL after 3 months and low of 1.01 ± 0.01 mg/mL after 6 months. Percentage of free dye was 1.3 ± 0.1% on release for durvalumab-680LT and was 1.9 ± 0.006% after 12 months of stability testing, with a high at 6 months of 2.1 ± 0.06%. No measurement was lower during the stability study than the first measurement of free dye. No aggregate formation was observed during the stability study of durvalumab-680LT. Target binding affinity stayed between 63 and 98% for durvalumab-680LT. Concentration after production was 0.98 ± 0.003 mg/mL for nivoumab-800CW and was 0.95 ± 0 mg/mL after 12 months of stability testing, with a high of 1.00 ± 0.006 mg/mL after 3 months and low of 0.97 ± 0.0006 mg/mL after 6 months. Percentage of free dye was 0.4 ± 0.09% on release for nivolumab-800CW and was 4.3 ± 0.06% after 12 months of stability testing. A lows was found after 1 month of 0.3 ± 0.1%. No measurement was lower during the stability study than the last measurement of free dye. Aggregate formation was only observed for nivolumab-800CW after 12 months of stability testing with 2.2 ± 0.06% of aggregates. Target binding affinity stayed between 60 and 89% for nivolumab-800CW. All other QC tests also stayed within the end of shelf-life specifications at 15–25 °C. This demonstrates the stability of both durvalumab-680LT and nivolumab-800CW for 12 months at both temperature ranges.

### 2.4. Toxicity Study

Single-dose intravenous administration of the tracer durvalumab-680LT to mice, at the doses of 9 mg/kg and 90 mg/kg, and IRDye 680LT, at the doses of 86 µg/kg and 860 µg/kg, did not result in mortalities. Body weight and food consumption were unaffected by the treatment at all doses tested for both dye and tracer (Appendix A). There were no changes related to the tracer or dye in the clinical pathology parameters, haematology, coagulation, clinical chemistry, urinalysis, terminal fasting body weights, organ weights, and organ weight to weight ratios (Appendix A). Grossly, bilateral blue discoloration of iliac lymph node and blue discoloration at injection site were observed at 90 mg/kg durvalumab-680LT in both sexes on day 2 and 15 without histopathologic correlation. The few incidences of inflammatory cell infiltration at the site of injection in control and treatment groups were considered as procedure-related and not related to test item administration.

## 3. Discussion

We successfully developed and produced durvalumab-680LT and nivolumab-800CW according to cGMP. To our knowledge this is the first time ICIs have been labelled with fluorescent dyes for first-in-human use in phase I/II clinical trials. Both tracers can now be combined in these trials, enabling multispectral fluorescent molecular imaging of the PD-1/PD-L1 axis. This is unique and of particular interest, as this is a novel approach to FMI, serving as a potential tool to overcome persistent challenges in ICI treatment. In this paper we describe the development and clinical translation of durvalumab-680LT and nivolumab-800CW.

Both tracers were developed using the “roadmap for development and clinical translation of 800CW optical tracers”, set up previously by our group [30]. Durvalumab was successfully conjugated to IRDye 680LT, whereas nivolumab was successfully conjugated to IRDye 800CW. Labelling optimization yielded parameters similar to previously developed tracers, including labelling time, label ratio, and used buffers, indicating that different antibodies display similar optimal labelling conditions and parameters [30,31,32]. Validation batches produced in our cGMP facility remained within all set specifications for 12 months at both 2–8 °C as 15–25 °C, indicating that the production process can be used to produce a stable tracer product fit for clinical production and subsequently for use in phase I/II clinical trials. Stability studies are ongoing and will collect data at 2–8 °C for up to 24 months to allow tracer use over a long shelf-life period.

Several fluorescent tracers have been developed previously, including bevacizumab-800CW, cetuximab-800CW, panitimumab-800CW, and vedolizumab-800CW; however, these tracers were all based on a therapeutic antibody conjugated with IRDye 800CW [30,31,32,33,34,35]. To enable multispectral fluorescent imaging of the PD-1/PD-L1 axis, we aimed to use IRDye 680LT in addition to IRDye 800CW. IRDye 800CW safety has been previously demonstrated, both unconjugated and conjugated to bevacizumab [31,36]. Moreover, multiple clinical trials have been performed with antibodies conjugated to this dye [25,34,37,38,39]. Therefore, no additional safety testing was performed for nivolumab-800CW. In contrast, IRDye 680LT has never been used in humans. Therefore, an animal toxicity study in CD-1 mice was included in the development of durvalumab-680LT, assessing both the toxicity of IRDye 680LT and IRDye 680LT conjugated to durvalumab. Study results showed no observed toxic effects from the conjugated antibody or the dye, similar to the previously performed toxicity studies for 800CW-based tracers [31,36], supporting the safe use of IRDye 680LT in humans. This work thus opens up possibilities for conjugation of IRDye 680LT to other antibodies or molecules to image new targets of interest using IRDye 680LT or multispectral imaging.

Several optical tracers are already being used in a clinical trial setting, which have proven to be able to distinguish tumour tissue from normal tissue [25,34,37,38,39]. However, no fluorescent ICI tracers have been previously used in clinical trials, making durvalumab-680LT and nivolumab-800CW the first ones to be used. In contrast, the PET-tracers ^89^Zr-atezolizumab, ^89^Zr-nivolumab, and ^89^Zr-pembrolizumab have been used before in clinical studies to assess ICI therapy response. For these tracers, correlations were found between tracer uptake in the tumour pretreatment and both response to therapy and (progression-free) survival [18,19,20]. In addition, a one-armed antibody targeting CD8 has also been labelled with ^89^Zr to characterize the dynamics of CD8^+^ T cells in the context of ICI treatment [40]. As CD8^+^ T-cell proliferation at baseline in the tumour area is associated with increased ICI therapy response and tumour reduction, these radiolabelled tracers are a great addition for patient stratification already. Moreover, fluorescent imaging can be a great complementary technique, especially when using multispectral imaging [41,42,43]. Whereas PET imaging can cover the whole body and has a high penetration depth, fluorescence imaging provides high-resolution and real-time visualization of the tumour in vivo during surgery or endoscopy, limiting the impact of the lower penetration depth of fluorescence imaging. Moreover, since fluorescent tracers do not have a radioactive half-life, ex vivo analysis can be performed to get a more mechanistic insight into drug distribution of the antibodies, yielding insight into the heterogeneity of the tumour on a cellular level, the penetration of effector cells into the tumour microenvironment and interactions at a molecular immunological level within the tumour [44,45]. Multispectral imaging can further enhance these insights and could give additional information on interactions between targets. Together with novel techniques like needle-based ultrasound guided quantitative fluorescence molecular imaging, these results can be linked to the immune composition of the tumour microenvironment giving additional insights into tumour heterogeneity and can thereby potentially help to better understand why certain patients respond to ICI therapy and others do not. This would potentially improve patient stratification, as well as enable treatment response assessment and response prediction.

Durvalumab-680LT is currently being used in a first-in-human study (NCT05450484) at the University Medical Center Groningen to assess the tumour heterogeneity of PD-L1 expression in oesophageal cancer before and after neoadjuvant chemoradiotherapy. Nivolumab-800CW is currently being prepared for use in a clinical trial setting, where it will be combined with durvalumab-680LT, enabling multispectral imaging of the immune checkpoint pathway in oesophageal cancer. Moreover, durvalumab-680LT and nivolumab-800CW will be combined for multispectral imaging in locally advanced rectal cancer (NCT06304597). Depending on the results of these first clinical studies, this multispectral imaging approach using durvalumab-680LT and nivolumab-800CW can be investigated in several forms of advanced and metastatic cancers.

## 4. Materials and Methods

### 4.1. Labelling Procedure

For the development of durvalumab-680LT we used the registered drug product Imfinzi^®^ (AstraZeneca, Cambridge, UK) and for the development of nivolumab-800CW we used the registered drug product Opdivo^®^ (Bristol-Myers Squibb, New York, NY, USA). As an initial test, both nivolumab and durvalumab were labelled with both IRDye 800CW and IRDye 680LT (both LI-COR biosciences Lincoln, NE, USA), yielding durvalumab-800CW, durvalumab-680LT, nivolumab-800CW, and nivolumab-680LT. Tracer production procedures were derived from the development procedures of bevacizumab-800CW, cetuximab-800CW, and trastuzumab-800CW [30,31]. Briefly, durvalumab or nivolumab were buffer exchanged to labelling buffer, a 50 mM sodium phosphate buffer pH 8.5 (Apotheek A15, Gorinchem, The Netherlands), using PD-10 desalting columns (Cytiva lifesciences, Chicago, IL, USA), which were pre-equilibrated with labelling buffer. Afterwards, IRDye 800CW or IRDye 680LT, both dissolved in DMSO (Merck, Darmstadt, Germany), were added to the protein solution in a molar dye to protein ratio of 2:1. The solution was mixed gently and left to incubate protected from light for 1–2 h at room temperature. After incubation, the solution was purified from any unconjugated dye by buffer exchange on PD-10 desalting columns using the optimal formulation buffer: a 50 mM sodium phosphate buffer at pH 7.0 (Apotheek A15, Gorinchem, The Netherlands). Afterwards, the tracer solution was diluted with the same phosphate buffer to a concentration of 1.0 mg/mL.

The tracers were analysed several times during the production process using SE-HPLC to assess the critical process parameters and results of conjugation. At the end of the conjugation reaction, the percentage of dye conjugated to the antibody relative to the total dye added, the label efficiency, was determined. After purification, a second analysis was performed to determine the tracer concentration, which was used to dilute the tracer to the target concentration of 1.0 mg/mL. Afterwards a final test was performed to determine concentration and the percentage of free unconjugated dye and protein aggregates, as well as tracer identity and integrity and yield of the tracer.

### 4.2. Labelling Optimization

Based on the initial labelling results, optimization of the labelling process was performed for durvalumab-680LT and nivolumab-800CW.

First, the influence of amino acids present in the original formulation of the antibodies on the labelling process was investigated. Labelling with a buffer exchange into a 50 mM sodium phosphate buffer pH 8.5, using PD-10 desalting columns pre-equilibrated with the same buffer, was compared to labelling without buffer exchange.

Second, the incubation time of the antibody–dye solution was assessed. Standard labelling protocol was followed. The beginning of the labelling process was the addition of the dye to the antibody solutions. Samples were taken at regular intervals of 20 min up to 120 min. LE was measured at all time points.

Third, the optimal molar label ratio between the antibody and the dye was assessed. Both tracers were labelled with ratios of 1:1, 2:1, 4:1, 8:1, and 16:1 (dye to protein) following standard labelling protocol. LE, protein concentration, and the percentage of aggregates and free dye were measured. Moreover, the fluorescence and absorbance for all label ratios was measured using a BioTek Synergy H4 plate reader (Winooski, VT, USA). Two concentrations of the tracer, 0.5 mg/mL and 0.1 mg/mL were added to a 96-well plate for all label ratios. Absorption spectra were measured in a range of 230 to 850 nm, whilst fluorescence intensity was measured with λ_ex_ of 700 nm and λ_em_ of 789 nm for the IRDye 800CW and with λ_ex_ of 632 nm and λ_em_ of 676 nm for the IRDye 680LT. Data were corrected for blank (unconjugated) antibody intensity.

Fourth, eight buffers for formulation of the final products were assessed by formulating both tracers in all different buffers after labelling following standard protocol. The composition of the different buffers, as well as the original buffers of both durvalumab and nivolumab, can be found in Appendix A. A buffer panel and a small-scale stability study (28 days) were set up for both tracers for all buffers. The formulation buffers tested were modified sodium phosphate buffers, partially based on the original formulation buffer of the antibodies.

### 4.3. cGMP Production, Validation, and Long-Term Stability

Using the optimized labelling conditions, the production process was translated to our cGMP facility to produce a full-scale clinical batch of both tracers. Before the first clinical production, a validation batch was produced, validating the production process and finalizing the product specifications before production of the first clinical batch. The produced vials were all used for a long-term stability study at two different temperatures according to the ICH Q1A guideline: 2–8 °C (refrigerated) for 24 months and 15–25 °C (room temperature) for 12 months. Stability time points and performed tests can be found in Appendix A.

cGMP production of durvalumab-680LT and nivolumab-800CW consisted of conjugation, purification, and sterile filtration. Briefly, durvalumab or nivolumab was buffer exchanged to labelling buffer, a 50 mM sodium phosphate buffer pH 8.5 using PD-10 desalting columns, which were pre-equilibrated with labelling buffer. Afterwards, cGMP-grade IRDye 800CW (nivolumab) or cGMP-grade IRDye 680LT (durvalumab), both dissolved in DMSO, was added to the protein solution in a molar dye to a protein ratio of 2:1. The solution was mixed gently and left to incubate protected from light for 1–2 h at room temperature. After incubation, the solution was purified from any unconjugated dye by buffer exchange on PD-10 desalting columns using the optimal formulation buffer: a 50 mM sodium phosphate buffer at pH 7.0 (Apotheek A15, Gorinchem, The Netherlands). Afterwards, the tracer solution was diluted with the same phosphate buffer to a concentration of 1.0 mg/mL. Subsequently, the solution was filtered over a sterile 0.2 µm filter (Sartopore^®^ 2 gamma filter capsule), which was connected to a Flexboy^®^ bag (3 L, Sartorius Stedim, Göttingen, Germany). Next, aseptic filling over a sterile 0.2 µm filter (Millex GP, Merck KGaA, Darmstadt, Germany) into 10R vials (APG Pharma, Uithoorn, The Netherlands) was performed, after which vials were closed using a 20 mm bromobutyl rubber stopper (Datwyler, Altdorf, Switzerland) and 20 mm aluminium closure (Datwyler, Altdorf, Switzerland). Closures were crimped using a compressed air-powered semi-automatic crimping tool.

A schematic overview of the whole development and optimization process, including the long-term stability testing and all relevant QC tests, is displayed in Figure 4.

### 4.4. Quality Control Testing

Product specifications of durvalumab-680LT and nivolumab-800CW at release and at end of shelf life are presented in Table 2. In addition to pharmacopeial specifications, specific product specifications are based on development and optimization data. All specifications are identical for both tracers, except for the tests that involve the absorption spectrum of the specific dye.

The tracers were analysed using SE-HPLC as described previously [30]. During analysis, the absorption spectrum was measured from 200 to 900 nm. Using this absorption spectrum, HPLC analysis for the IRDye 800CW and IRDye 680LT was performed. The absorbance spectra of peaks from conjugated antibody and unconjugated dye were analysed to determine the absorption peak of the dye before and after conjugation. For protein measurements, peaks were analysed at 280 nm. A wavelength of 780 nm was used for the analysis of IRDye 800CW conjugated to antibodies and 775 nm was used for the analysis of unconjugated IRDye 800CW. For IRDye 680LT, 679 nm was used for dye conjugated to an antibody and 676 nm was used to determine the unconjugated dye.

Target affinity was assessed by an in-house developed indirect ELISA. ELISA set-up and design was performed as described in previous protocols [30]. Individual assay development and optimization was performed for both nivolumab-800CW and durvalumab-680LT. Briefly, a 96-well plate was coated overnight with recombinant human PD-L1 (Sino Biologicals Europe GmbH, Eschborn, Germany) (0.5 µg/mL) at 2–8 °C for durvalumab-680LT or recombinant human PD-1 (Sino Biologicals Europe GmbH, Eschborn, Germany) (0.5 µg/mL) at 2–8 °C for nivolumab-800CW. The next day, the plate was blocked with 1% bovine serum albumin (VWR international BV, Amsterdam, The Netherlands). Dilution series of both the commercially available durvalumab and durvalumab-680LT (10 µg/mL to 10 ng/mL) or of both the commercially available nivolumab and nivolumab-800CW (1.5 µg/mL to 1.5 ng/mL) were made and incubated on the plate in duplicate. Bound antibody was detected using an anti-human IgG secondary antibody labelled with horseradish peroxidase (HRP) (Jackson Immunoresearch laboratories, West Grove, PA, USA), specific against the Fab part of human IgG. Signal was generated by addition of 3,3′,5,5′-tetramethylbenzidine (TMB, Thermo Fisher, Waltham, MA, USA) and the reaction was stopped with 2 M sulfuric acid. TMB absorption was read at 450 nm. A four-parameter logistic regression was performed on the data to fit a sigmoidal curve. From the sigmoidal fit, the EC_50_ value was calculated. Target affinity was calculated by comparing the EC_50_ of the tracer and the reference commercially available antibody, yielding the relative binding affinity of the tracer.

The general appearance, turbidity, visible particles, and colour of both tracers were assessed visually. Bacterial endotoxins, bioburden, residual solvents (DMSO), osmolality, pH, and sub-visible particles were tested according to their respective compendial Ph. Eur. monographs [46].

For both tracers, ELISA and HPLC methods were validated according to ICH Q2 guidelines. All other tests were validated according to their respective Ph. Eur. monographs [46].

### 4.5. Toxicity Study

An extended single-dose toxicity study in CD-1 mice was conducted according to the ICH M3 R2 guidelines. Moreover, in accordance with FDA guideline exploratory IND studies, a low dose (100× intended clinical study dose, based on microdosing of 30 nmol) and a high dose (1000× intended clinical study dose) of both the tracer durvalumab-680LT and the dye IRDye 680LT were administered. This corresponds with a single intravenous bolus injection of durvalumab-680LT (9 or 90 mg/kg body weight) or IRDye 680LT (86 or 860 µg/kg body weight). The study was conducted by Eurofins Advinus Ltd. (Bengaluru, India). The study protocol was reviewed and approved by the Institutional Animal Ethics Committee (IAEC), which is equivalent to Institutional Animal Care and Use Committee (IACUC). All procedures were in compliance with the guidelines issued by the Committee for the Purpose of Control and Supervision of Experiments on Animals (CPCSEA), India.

This toxicity study was set up to assess the toxic potential of both durvalumab-680LT and unconjugated IRDye 680LT when administered as a single intravenous dose to CD-1 mice and to study the reversibility of possible effects. No cross-reactivity between durvalumab and mouse PD-L1 was expected [47]. The mice, CRL:CD-1 (ICR), were obtained from HyLasco Biotechnology (India) and breeders were originally obtained from Charles River (Sulzfeld, Germany). At the commencement of treatment, on average, males weighed 33.26 g and females 26.82 g. Weight variation of mice did not exceed ±20% of the mean body weight in each sex and group. Females were nulliparous and non-pregnant.

Mice were assigned to groups, as shown in Table 3. Randomization of the mice was performed using a computer-generated algorithm according to body weight. The mice received a single intravenous bolus injection of durvalumab-680LT (9 or 90 mg/kg body weight), IRDye 680LT (86 or 860 µg/kg body weight), or vehicle (saline) via the tail vein. Ten mice of each sex were sacrificed two days after injection; the remaining mice were observed during a two-week follow-up and were sacrificed on day 15. At the time of administration, the animals were approximately 6 weeks old. The mice were observed at least twice daily for morbidity and mortality. Moreover, changes in skin, fur, eyes, mucous membranes, excretion, secretion and autonomic activity, body weight, and food consumption were measured at weekly intervals after administration. After sacrificing of the mice, organ and tissue weight were determined, as well as changes in haematology, clinical chemistry, and histopathology. The Provantis^TM^ software (Instem, Stone, Sheffordshire, UK) was used to collect all of this data.

### 4.6. Statistical Analysis

Data captured with Provantis^TM^ were analysed using built-in statistical analysis. A decision tree was used in analysis. First, test variance homogeneity was tested by Levene’s method. When variances were heterogenous, suitable transformation was performed automatically by the software. Second, further one-way analysis of variance (ANOVA) was performed. If ANOVA was significant, Dunnett’s vehicle control versus treatment group mean comparison was performed. Thirdly, for two groups, the comparison of mean between treatment and control group was performed using student’s *t*-test. For SE-HPLC analysis, mean values and standard deviations were calculated unless stated otherwise. For other QC tests, means were calculated where applicable.

## 5. Conclusions

In conclusion, in this study we developed the fluorescent tracers durvalumab-680LT, targeting PD-L1, and nivolumab-800CW, targeting PD-1, which are suitable for human use and can be applied in phase I/II clinical trials. Moreover, we have shown that both IRDye 680LT and durvalumab-680LT showed no toxic effects and are therefore safe for in-human use. Together, these ICI tracers enable a multispectral imaging approach on the PD-1/PD-L1 axis, giving insight into heterogeneity of the tumour and interactions between targets. Ultimately, this may be a major step in improving ICI patient stratification, enabling better response prediction or response assessment and ultimately improving ICI treatment.

## Figures and Tables

**Figure 1 pharmaceuticals-18-01501-f001:**
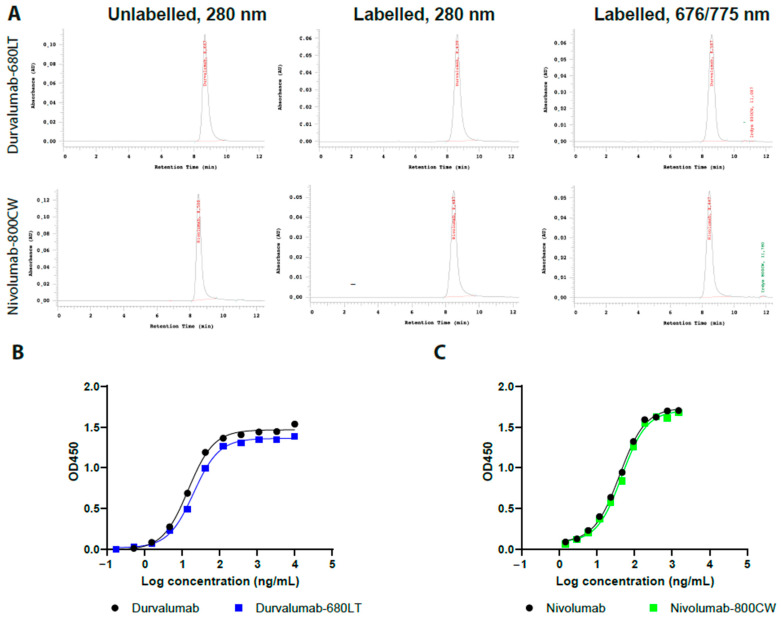
(**A**) Representative size-exclusion high-performance liquid chromatography (SE-HPLC) chromatograms of durvalumab-680LT and nivolumab-800CW. First column represents both unconjugated antibodies at 280 nm. Here, the main peak of the antibody can be seen around the 9 min mark. Second column represents both antibodies conjugated to their respective dye at 280 nm. Here the main peak of the antibody–dye conjugate can be seen around the 9 min mark, whilst any aggregates would be visible before the main peak. Third column represents durvalumab-680LT at 676 nm and nivolumab-800CW at 775 nm, confirming successful conjugation of the dye to the protein. Here, the main peak of the antibody can be seen around the 9 min mark, whilst any aggregates would be visible before the main peak. Peaks for the free dye can be seen around the 12 min mark. Representative results of the indirect ELISA for (**B**) durvalumab-680LT and (**C**) nivolumab-800CW. (**A**–**C**) are single measurements, so no statistics were performed.

**Figure 2 pharmaceuticals-18-01501-f002:**
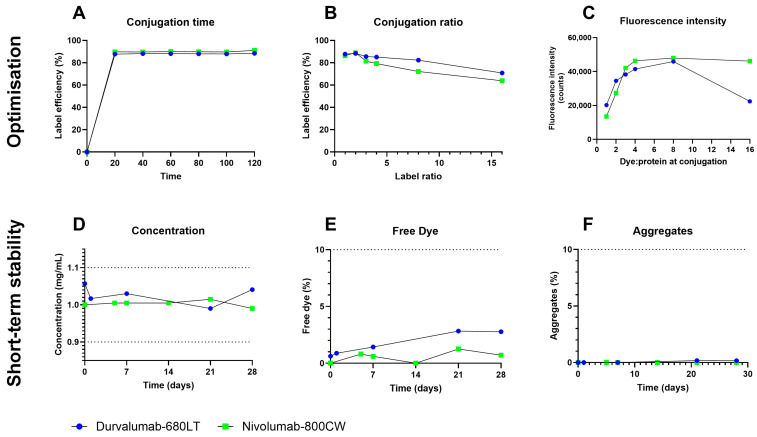
Durvalumab-680LT and nivolumab-800CW conjugation optimization and small-scale stability. Optimization of the production process was achieved by investigating the optimum in conjugation time (**A**), the label efficiency at increasing amounts of dye per antibody (**B**), and the resulting fluorescent intensity of the signal after conjugation of higher amounts of dye (**C**). Optimized tracers were tested for preliminary stability by determining concentration of the tracer monomer (**D**) and presence of primary impurities-free IRDye (**E**) and antibody aggregates (**F**). Dotted lines in panels (**D**–**F**) indicate acceptance limits during stability study. All optimization experiments were only repeated once; therefore, no statistics were performed here.

**Figure 3 pharmaceuticals-18-01501-f003:**
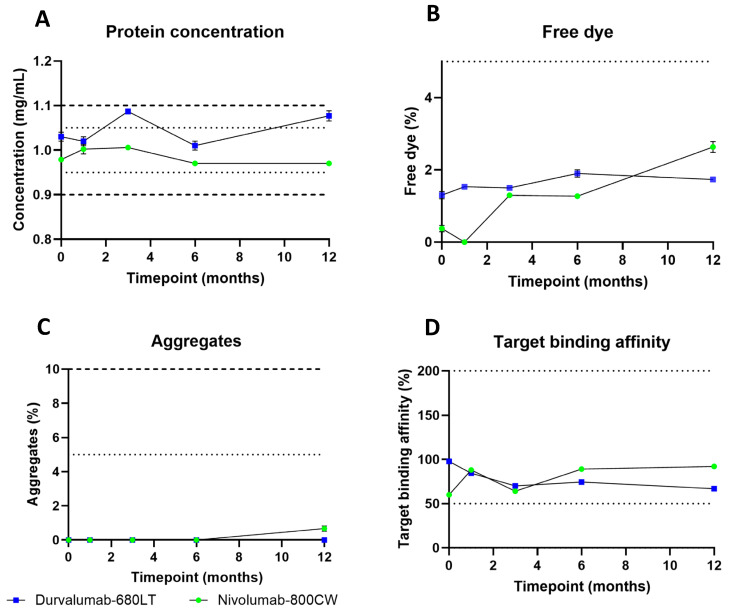
Stability results of durvalumab-680LT and nivolumab-800CW at 2–8 °C. Release specifications are displayed with dotted lines and end of shelf-life specifications are displayed with dashed lines. (**A**) Protein concentration, (**B**) percentage of free dye, (**C**) percentage of aggregates, and (**D**) target binding affinity of both tracers. (**A**–**C**) are means ± standard deviation of three different measurements, and in (**D**) mean is reported of two measurements.

**Figure 4 pharmaceuticals-18-01501-f004:**
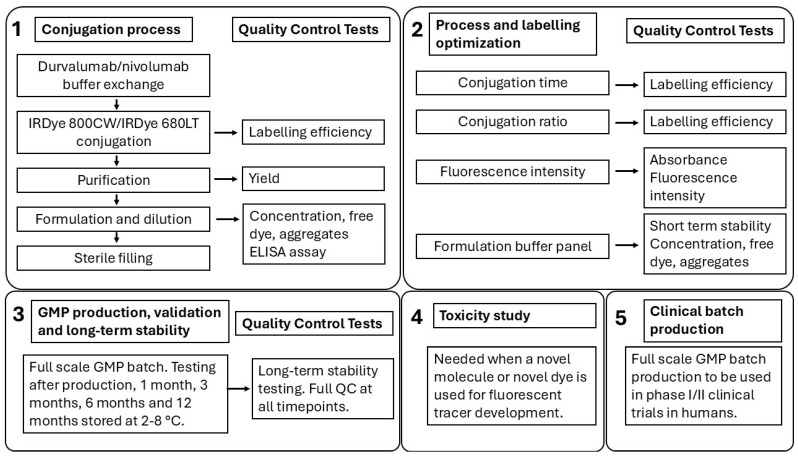
A schematic overview of the development and optimization process of fluorescent tracer development with all relevant QC tests.

**Table 1 pharmaceuticals-18-01501-t001:** Initial labelling results for durvalumab-800CW, durvalumab-680LT, nivolumab-800CW, and nivolumab-680LT, comprised of three different labelling experiments per tracer. Mean ± standard deviation is shown where applicable.

Tracer	Label Efficiency	Protein Yield	Concentration Sufficient	Protein Aggregates	Unconjugated Dye
Durvalumab-800CW	85.2 ± 1.1%	90.1 ± 2.3%	Yes	N.D.	0.38 ± 0.57%
Durvalumab-680LT	88.4 ± 2.0%	98.6 ± 2.5%	Yes	N.D.	0.48 ± 0.39%
Nivolumab-800CW	89.9 ± 0.5%	98.2 ± 2.7%	Yes	N.D.	0.50 ± 0.20%
Nivolumab-680LT	77.8 ± 3.0%	91.9 ± 1.7%	Yes	N.D.	2.0 ± 2.5%

**Table 2 pharmaceuticals-18-01501-t002:** Product specifications for durvalumab-680LT and nivolumab-800CW (both 1.0 mg/mL, 5.0 mL) at release and end of shelf life.

Test	Method	Release Specification	End of Shelf-Life Specification
**Protein monomer concentration**	SE-HPLC	0.95–1.05 mg/mL	0.90–1.10 mg/mL
**Protein aggregates**	SE-HPLC	≤5.0%	≤10%
**Unconjugated IRDye 800CW/680LT**	SE-HPLC	≤5.0%	≤10%
**Protein monomer identity**	SE-HPLC	Retention time comparable to reference standard	Retention time comparable to reference standard
**Protein monomer integrity**	SE-HPLC	Peak shape comparable to reference standard; no shoulders, minimal tailing	Peak shape comparable to reference standard; no shoulders, minimal tailing
**UV-VIS absorption peaks**	SE-HPLC	Nivolumab-800CW: peaks at 280 ± 3 nm 775 ± 3 nm Durvalumab-680LT: peaks at 280 ± 3 nm and 679 ± 3 nm	Nivolumab-800CW: peaks at 280 ± 3 nm 775 ± 3 nm Durvalumab-680LT: peaks at 280 ± 3 nm and 679 ± 3 nm
**Target binding affinity**	Indirect ELISA	50–200%	50–200%
**Appearance (turbidity)**	Visual inspection	Clear to slightly opalescent solution	Clear to slightly opalescent solution
**Appearance (colour)**	Visual comparison to colour standards	Colour tone: comparable to reference	Colour tone: comparable to reference
**Container closure and label**	Visual inspection	Closure intact, label legible and intact	Closure intact, label legible and intact
**Extractable volume**	Ph. Eur. 2.9.17	≥5.0 mL	≥5.0 mL
**pH**	Ph. Eur. 2.2.3	6.9–7.1	6.9–7.1
**Osmolality**	Ph. Eur. 2.2.35	270–310 mOsmol/kg	270–310 mOsmol/kg
**Residual solvents (DMSO)**	Ph. Eur. 2.4.24 Ph. Eur. 5.4	≤50.0 mg/L	≤50.0 mg/L
**Bacterial endotoxins**	Ph. Eur. 2.6.14	≤5.0 EU/mL	≤5.0 EU/mL
**Sterility**	Ph. Eur. 2.6.1	Sterile	Sterile
**Visible Particles**	Ph. Eur. 2.9.20	Practically free of visible particles	Practically free of visible particles
**Sub-visible Particles**	Ph. Eur. 2.9.19	Particles ≥10 μm: ≤6000/vial Particles ≥25 μm: ≤600/vial	Particles ≥10 μm: ≤6000/vial Particles ≥25 μm: ≤600/vial

**Table 3 pharmaceuticals-18-01501-t003:** Design of the toxicity study for durvalumab-680LT and IRDye 680LT.

Group	Treatment	Dose	No. of Mice *	Sex
G1	Vehicle control (saline)	N.A.	16	M
			16	F
G2	Durvalumab-680LT	9 mg/kg body weight	10	M
			10	F
G3		90 mg/kg body weight	16	M
			16	F
G4	IRDye 680LT	86 µg/kg body weight	10	M
			10	F
G5		860 µg/kg body weight	16	M
			16	F

* First 10 mice per group were sacrificed on day 2. The remaining mice were sacrificed on day 15. Abbreviations: N.A.: not applicable, M: male, F: female.

## Data Availability

The original contributions presented in this study are included in the article/Appendix A. Further inquiries can be directed to the corresponding author(s).

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
