# Peer review of "Development of Clinical-Grade Durvalumab-680LT and Nivolumab-800CW for Multispectral Fluorescent Imaging of the PD-1/PD-L1 Axis of the Immune Checkpoint Pathway"

_pharmaceuticals, 2025, doi:10.3390/ph18101501_

Round 1
Reviewer 1 Report
Comments and Suggestions for Authors
The authors in the manuscript entitled “Development of clinical grade durvalumab-680LT and nivolumab-800CW for multispectral fluorescent imaging of the PD-1/PD-L1 axis of the immune checkpoint pathway” presents a well-executed and commendable study. The authors developed conjugates of the anti-PD-L1 antibody durvalumab with IRDye 680LT, and the anti-PD-1 antibody nivolumab with IRDye 800CW. They conducted thorough stability and toxicity assessments in mice for the durvalumab-680LT conjugate. Overall, the work addresses a significant and timely topic in multispectral fluorescence imaging, contributing valuable insights to the field. I would recommend the acceptance of the manuscript after minor revisions, particularly addressing clarity on the below mentions points:
- The NIR imaging is limited by tissue penetration, the study does not fully address how this constraint may impact detection sensitivity of Durvalumab-680LT and nivolumab-800CW conjugation in humans. It would be useful to include a discussion on potential strategies to overcome imaging depth limitations in clinical applications.
- The authors have not mentioned how have they selected the doses for toxicity studies.
- Have the authors considered performing multispectral fluorescence imaging in mice using both the conjugates simultaneously?
Overall, the manuscript marks a significant contribution, and with the inclusion of these additional dimensions, it could serve as an even more impactful guide for future implementation efforts.
Reviewer 2 Report
Comments and Suggestions for Authors
The authors present a well-structured study with sound methodology and thoughtful interpretation. In this work, two immune checkpoint inhibitors—durvalumab (targeting PD-L1) and nivolumab (targeting PD-1)—were successfully conjugated to distinct near-infrared dyes, IRDye 680LT and IRDye 800CW, respectively. The resulting fluorescent tracers were systematically optimized to ensure high conjugation efficiency and purity, with the goal of advancing them toward clinical application. Stability assessments confirmed that both conjugates remained within acceptable specifications for up to 12 months under refrigerated conditions. Furthermore, a comprehensive single-dose toxicity evaluation in murine models demonstrated the safety of both the unconjugated dye and the durvalumab-IRDye 680LT conjugate at doses far exceeding those intended for human use. These findings support the translational readiness of both tracers, positioning them as viable candidates for early-phase clinical trials. The manuscript offers novel insights into the development and preclinical validation of fluorescently labeled immune checkpoint inhibitors, with implications for future imaging-guided therapeutic strategies. Nevertheless, there are several areas where the manuscript could be further improved to enhance its quality and impact. The authors are advised to revise the manuscript based on the following points:
1) To enhance the clarity and reproducibility of the manuscript, I strongly recommend the inclusion of a schematic overview summarizing the production workflow and quality control procedures for the fluorescently labeled immune checkpoint inhibitors. A visual representation outlining key steps—such as antibody-dye conjugation, purification, labeling efficiency assessment, stability testing, and toxicity evaluation—would greatly benefit readers by providing a concise and intuitive understanding of the experimental pipeline
2) The ELISA results should be described with more quantitative detail, including EC50 values or relative binding affinities, to substantiate the conclusion that tracer-target interactions remain intact post-conjugation
3) The stability study results are promising, but the narrative would benefit from clearer statistical framing. Indicate whether replicate measurements were performed, and consider reporting mean ± SD values for tracer concentration, free dye percentage, and aggregate formation. This will support the claim that the tracers remained within shelf-life specifications and reinforce their translational readiness
4) Although an extended single-dose toxicity study in CD-1 mice was conducted in accordance with ICH M3(R2) guidelines, the manuscript lacks preclinical data demonstrating the application of these immune checkpoint inhibitors (ICIs) in tumor-bearing mice using fluorescence molecular imaging. Including such data would significantly strengthen the translational relevance of the tracers and provide critical insight into their in vivo targeting performance and imaging potential in a disease-relevant model.
